# New Advances in Rapid Pretreatment for Small Dense LDL Cholesterol Measurement Using Shear Horizontal Surface Acoustic Wave (SH-SAW) Technology

**DOI:** 10.3390/ijms25021044

**Published:** 2024-01-15

**Authors:** Tai-Hua Chou, Chia-Hsuan Cheng, Chi-Jen Lo, Guang-Huar Young, Szu-Heng Liu, Robert Y-L Wang

**Affiliations:** 1Biotechnology Industry Master and PhD Program, Chang Gung University, Taoyuan 33302, Taiwan; achou2@mmm.com (T.-H.C.); youngguanghuar@gmail.com (G.-H.Y.); 2Graduate School of Science and Technology, Shizuoka University, 3-5-1 Johoku, Naka-ku, Hamamatsu-shi 432-8561, Japan; joshcheng@tst.bio; 3tst Biomedical Electronics Co., Ltd., Taoyuan 324403, Taiwan; 4Metabolomics Core Laboratory, Healthy Aging Research Center, Chang Gung University, Taoyuan 33302, Taiwan; chijenlo@mail.cgu.edu.tw; 5Clinical Metabolomics Core Laboratory, Chang Gung Memorial Hospital, Taoyuan 33302, Taiwan; 6Department of Biomedical Sciences, College of Medicine, Chang Gung University, Taoyuan 33302, Taiwan; 7Division of Pediatric Infectious Diseases, Department of Pediatrics, Chang Gung Memorial and Children’s Hospital, Linkou 33305, Taiwan; 8Kidney Research Center and Department of Nephrology, Chang Gung Memorial Hospital, Linkou 33305, Taiwan

**Keywords:** small dense LDL, apolipoprotein B, shear horizontal surface acoustic wave SH-SAW

## Abstract

Atherosclerosis is an inflammatory disease of the arteries associated with alterations in lipid and other metabolism and is a major cause of cardiovascular disease (CVD). LDL consists of several subclasses with different sizes, densities, and physicochemical compositions. Small dense LDL (sd-LDL) is a subclass of LDL. There is growing evidence that sd-LDL-C is associated with CVD risk, metabolic dysregulation, and several pathophysiological processes. In this study, we present a straightforward membrane device filtration method that can be performed with simple laboratory methods to directly determine sd-LDL in serum without the need for specialized equipment. The method consists of three steps: first, the precipitation of lipoproteins with magnesium harpin; second, the collection of effluent from a 100 nm filter; and third, the quantification of sd-LDL-ApoB in the effluent with an SH-SAW biosensor. There was a good correlation between ApoB values obtained using the centrifugation (y = 1.0411x + 12.96, r = 0.82, n = 20) and filtration (y = 1.0633x + 15.13, r = 0.88, n = 20) methods and commercially available sd-LDL-C assay values. In addition to the filtrate method, there was also a close correlation between sd-LDL-C and ELISA assay values (y = 1.0483x − 4489, r = 0.88, n = 20). The filtration treatment method also showed a high correlation with LDL subfractions and NMR spectra ApoB measurements (y = 2.4846x + 4.637, r = 0.89, n = 20). The presence of sd-LDL-ApoB in the effluent was also confirmed by ELISA assay. These results suggest that this filtration method is a simple and promising pretreatment for use with the SH-SAW biosensor as a rapid in vitro diagnostic (IVD) method for predicting sd-LDL concentrations. Overall, we propose a very sensitive and specific SH-SAW biosensor with the ApoB antibody in its sensitive region to monitor sd-LDL levels by employing a simple delay-time phase shifted SH-SAW device. In conclusion, based on the demonstration of our study, the SH-SAW biosensor could be a strong candidate for the future measurement of sd-LDL.

## 1. Introduction

Cardiovascular diseases (CVDs) are the leading cause of death worldwide, killing an estimated 17.9 million people annually. CVDs are a group of diseases that affect the heart and blood vessels. They include coronary heart disease, cerebrovascular disease, rheumatic heart disease, and other diseases. People in low- and middle-income countries with CVDs and other noncommunicable diseases are less likely to have access to efficient and equitable health care services that meet their needs [1].

It is well known that low-density lipoprotein cholesterol (LDL-C) is one of the major risk factors for coronary artery disease (CAD) [2,3,4,5,6,7]. Although LDL-C is an important risk factor for CADs, LDL-C levels are not always elevated in patients with CAD [8]. There is growing evidence that the predominance of small dense LDLs (sd-LDLs) is strongly associated with CAD. In patients with type 2 diabetes and metabolic syndrome, sd-LDL-C concentrations are elevated in those at high risk for CAD [9,10,11,12]. Importantly, elevated sd-LDL-C is thought to be a feature of familial combined hyperlipidemia, a phenomenon involving hypercholesterolemia and/or hypertriglyceridemia, which is associated with an increased risk of premature CVD. Recent studies have shown that sd-LDL-C is associated with CVD risk, metabolic dysregulation, and multiple pathophysiological processes [13,14,15]. Therefore, the concentration of sd-LDL-C reveals the role of this lipoprotein subclass in cardiovascular pathophysiology and makes it available to physicians as a complementary biomarker.

LDL subclasses can be isolated by various laboratory procedures [16]. A range of methods is available for analyzing LDL subclasses and measuring sd-LDL particle size and number and cholesterol concentrations, such as ultracentrifugation (UC), vertical auto profile, gradient gel electrophoresis (GGE), nuclear magnetic resonance (NMR) spectroscopy, high performance liquid chromatography (HPLC), and ion mobility analysis(IM) [17]. Recently, a detergent-based method for sd-LDL-C homogeneity has been developed [18]. This method does not require any pretreatment and the has a high reproducibility of measurements using an automated analyzer. The current sd-LDL technique is flawed and produces inconsistent results, limiting its application in research laboratories. None of the currently proposed formulas for measuring sd-LDL have been properly validated for use as a clinical tool. To further reduce risk, studies of the particle size, composition, and amount of sd-LDL in various clinical situations have the potential to further our understanding of the atherosclerotic process and to provide information on risk stratification for CVD beyond LDL-C. However, many analytical techniques are available for the detection of sd-LDL particles. Their use is still largely limited to research laboratories because their analytical and clinical performance and the clinical and cost effectiveness of sd-LDL assays have not been fully demonstrated.

Although sd-LDL-C is strongly associated with CAD, few conclusive studies have shown that lowering sd-LDL-C reduces the risk of coronary artery disease [19,20,21]. Future clinical studies need to answer this question using accurate and reproducible sd-LDL-C measurement techniques. Thanks to intervention studies using robust statins and standardized sd-LDL-C measurements, these questions should soon be answered.

The fundamental principle of surface acoustic wave (SAW) sensors is rooted in the notion that the binding of biomolecules to the functionalized sensing layer induces an increase in mass and disturbance in the viscoelastic characteristics. Consequently, this induces a change in frequency or phase, which can be quantified through the utilization of a frequency counter or network analyzer [22].

In our investigation, we employed a shear horizontal surface acoustic wave (SH-SAW) biosensor incorporating an interdigital transducer (IDT) for both input and output, a sensing region coated with antibodies or antigens specific to the target molecules, and a reflector. Upon initiation of the measurement process, the initial electrical signal is converted into surface acoustic waves by the input IDT. These waves propagate along the sensing area and are subsequently reflected through the reflector to the output IDT. Ultimately, the waves are reconverted into electrical signals. The output signal corresponds to the binding events occurring within the sensing region. As the target molecules bind to the sensing area, the velocity and amplitude of the waves decrease. The magnitude of the signal generated is directly proportional to the extent of binding between the target protein or antibody and the sensing region [23,24,25].

It is well known that precipitation of apolipoprotein B (ApoB)-containing lipoproteins with a mixture of divalent cations and polyanions allows determination of HDL cholesterol [26,27]. Previous studies have shown that the small dense fraction of LDL remained in the supernatant, suggesting that not all ApoB-containing lipoproteins can be precipitated by the combination of heparin and magnesium [28]. Here, we present a straightforward membrane filtration method that allows for the direct determination of sd-LDL in plasma using a simple laboratory method without the need for specialized equipment. In this study, our preprocessing method is easily adaptable to screening, clinical testing, and other uses that require the rapid evaluation of large numbers of samples. In addition, we present an extremely sensitive and specific shear horizontal surface acoustic wave (SH-SAW) biosensor with apolipoprotein B antibodies in its sensitive region to monitor sd-LDL levels by employing a simple delay-time phase shifted SH-SAW device. The SH-SAW biosensor does not require a washing process or a complex flow system. Due to the wash-free, rapid, and quantitative immunoassay, the SH-SAW biosensor promises to be the most advanced point-of-care testing (POCT) application. We developed a simple pretreatment method and rapid SH-SAW ApoB measurement compared to current commercial sd-LDL assays. Based on our results, the SH-SAW biosensor may be a strong candidate for measuring sd-LDL in the future.

## 2. Results

### 2.1. Measurements of Standard Curves for SH-SAW Biosensor

The ApoB Calibrator (Beckman Coulter, Ireland) was purchased to establish standard curves. First, the apolipoprotein B calibrators were resuspended to 24, 48, 105, and 148 mg/dL with PBS solution as 0 mg/dL. The samples were diluted 20-fold before measuring the SH-SAW ApoB biosensor. To establish the standard curve, the 30-s phase shift endpoint was measured as the SAW signals for different concentrations of total ApoB, as shown in Figure 1a, with the following four-parameter logistics (4PL) equation (Figure 1b):
SAW signal = a + (b − a)/{1 + ([A ApoB concentration]/c)^d}
where a = −0.46877 b = 2.00127, c = 75.14937, and d = −30.90905 are the coefficients for the SAW biosensor chip with a coefficient of correlation (R) of 0.9962.

### 2.2. Comparison of sd-LDL-apoB by SH-SAW Biosensor under Different Pretreatment Methods

The principle of this method offers the unique possibility of precipitating lipoproteins in an in vitro system using only physiological polyanions such as heparin and magnesium. In this study, different filtration grades were tested to evaluate the precipitation effect of precipitation modifiers. Figure 2 shows the results of filtration efficiency for different pore size classes. It is noteworthy that a filter with a pore size of 300 kDa removed most of the ApoB proteins and showed very little phase shift on the SH-SAW biosensor. As the pore size of the filter increases, the 100 nm filter comes very close to the sd-LDL-ApoB results of current technology. In short, the 100 nm filter may be the optimal filtration grade for precipitating sd-LDL-ApoB using the SH-SAW biosensor.

Next, the amount of apoB proteins in the effluent was also determined using NMR spectroscopy to further investigate the filtration capacity. Table 1 shows that the study target was LDL-6 ApoB (1.044–1.063 g/mL) [29], with sd-LDL-ApoB of 14.8 mg/dL. NMR spectra measurements of the 100 nm filter effluent showed the best match to the study target LDL-6 ApoB values. Based on this result, we concluded that the 100 nm filter should be the best choice for the next phase of testing on a large number of specimens.

### 2.3. Comparative Study between Commercial sd-LDL-Cholesterol Assay Results and sd-LDL-ApoB Results from the SH-SAW Biosensor in a Large Study Group

Next, the above method was applied to measure plasma samples from patients. The results show a resolvable correlation between the commercial value of the sd-LDL-C kit and the SH-SAW value of sd-LDL-ApoB from the heparin-Mg^2+^ supernatant in 20 individuals (y = 1.0411x + 12.96, r = 0.82, n = 20) (Figure 3a). This suggests that the SH-SAW biosensor is capable of estimating sd-LDL-ApoB in conjunction with heparin-Mg^2+^ precipitates in a subsequent centrifugation step. However, we believe that the coefficients of the regression results are not as good as initially expected, because the collection of supernatants in routine clinical applications may require crude human manipulation and a longer centrifugation process. To further study this, we measured the sd-LDL-ApoB content in the filtrate with an SH-SAW biosensor using heparin-Mg^2+^ as precipitant and a 100 nm membrane filter. There was also a good correlation between the commercial sd-LDL-C assay and the sd-LDL ApoB measured by the SH-SAW ApoB biosensor (y = 1.0633x + 15.13, r = 0.88, n = 20) (Figure 3b).

Altogether, the filtration method not only had a shorter pretreatment time, but also had a more reliable effluent than the supernatant collection method. Most importantly, the filtration method showed better regression coefficients compared with the commercial sd-LDL-C assay.

### 2.4. Comparative Study of a Commercial of sd-LDL-ApoB Kit and an Apolipoprotein B-100 ELISA Kit

To confirm the accuracy of this method, filtration pretreatment is more convenient and ready than centrifugal separation. The filtered effluent was determined using an ApoB-100 ELISA kit and compared with the commercial sd-LDL assay. The reason for using the ApoB-100 ELISA kit is that Apo B-100 is produced by the liver and is a component of several other lipoproteins. Specifically, this protein is a component of very low-density lipoproteins (VLDLs) and low-density lipoproteins (LDLs). As shown in Figure 4, there was a very close correlation between sd-LDL-C and ELISA assay values in the filtrate assay (y = 1.0483x − 4489, r = 0.88, n = 20). After pretreatment with heparin-magnesium precipitant, the effluent showed a strong correlation with the sd-LDL-C results. This indicates that the 100 nm filter removes most of the non-sd-LDL ApoB, while only the sd-LDL-ApoB remain in the effluent.

### 2.5. Comparison Study between LDL Subfractions LDL-6 Apo-B, and sd-LDL-ApoB Results by SH-SAW Biosensor in a Large Study Group

To further confirm the accuracy of this method, the correlation between filtration pretreatment and NMR spectra was investigated. The sd-LDL-ApoB obtained from filtrate was determined with an SH-SAW biosensor using heparin-Mg^2+^ as the precipitant and a 100 nm membrane filter. Similarly, good correlations were shown between the LDL subfractions, Apo-B, LDL-6 and the sd-LDL ApoB measured with the SH-SAW ApoB biosensor (y = 2.4846x + 4.637, r = 0.89, n = 20) (Figure 5). Our study confirmed the filtration pretreatment combined with SH-SAW biosensor show a great ability to measure sd-LDL ApoB. NMR is indeed a good tool for analyzing sd-LDL or sd-LDL-C, but the overall equipment is expensive, and the operation requires professional personnel for daily maintenance or analysis, which may not be suitable for commercial clinical analysis In comparison, if the SH-SAW ApoB chip analysis platform is used, it has the advantages of miniaturization, fast analysis times, etc., and it has good consistency with the results of NMR analysis. We believe that in the future, the analysis of sd-LDL and the clinical application of LDL-ApoB will have great development potential and market space.

## 3. Discussion

The aim of this study was to develop a novel SH-SAW-based biosensor combined with an easy-to-operate pretreatment method to measure sd-LDL-ApoB in human plasma using SH-SAW technology. Since lipoprotein separation is not always precise, different lipoprotein particles, such LDL and Lp(a) particles, may show up in the same fraction or subfraction, depending on the method [18]. Although the distribution of LDL particles and their characteristics can be described by comparable nomenclature, it is debatable whether different methods define particles in the same way. There is an urgent need for a rapid and reliable method to detect sd-LDL-C. In the study, a novel sample pre-treatment method was introduced, utilizing a precipitation agent consisting of heparin and Mg^2+^. This method facilitated the formation of larger particle-like precipitates of large, buoyant LDL-C, which were then separated using a 100 nm filtration membrane. The separated filtrate was subsequently measured for the concentration of ApoB using the SH-SAW A biosensor. From the experimental results, a strong positive correlation was observed with the results obtained from the commercially available sd-LDL-C reagent analysis, as well as the NMR spectroscopy and ELISA kit results. Therefore, it is believed that this is a promising new clinical diagnostic tool for the rapid measurement of sd-LDL-C.

From the results of our experiments, we once again confirmed that sd-LDL and large buoyant LDL can be successfully separated using a simple reagent containing heparin and magnesium chloride, and further in combination with other analytical methods, sd-LDL-ApoB can be effectively performed and the concentrations of sd-LDL-C in the samples can be accurately determined. However, the effectiveness of the pre-treatment method will directly affect the accuracy and efficiency of the subsequent analytical instruments. In our experimental results, the SH-SAW biosensor was successfully used to find out the most suitable filter pore size by using different filter suppliers with different filter pore sizes. We also found that analysis of the filtered filtrate by conventional ELISA ApoB 100 assay also correlated well with commercial results. The above experimental results show that we can use the simple, inexpensive pretreatment reagents of heparin and magnesium chloride and good filters without the need for more expensive commercial chemicals, and that it can be a very efficient and highly accurate pretreatment method. In this study, we also found that the experimental results showed that the detection of sd-LDL-ApoB in the pretreated samples using the conventional ELISA ApoB 100 assay and the SH-SAW biosensor showed a good linear correlation after the same pretreated samples, as shown in Figure 6 (y = 0.834x + 25.479, r = 0.88, n = 20). Therefore, we can replace ELISA assay with the SH-SAW biosensor.

The SH-SAW technique is designed and constructed with an inter-digital transducer that generates acoustic waves on a solid surface, allowing us to directly observe changes in the wave characteristics of a specific marker as it binds to the SH-SAW surface [30]. Although the SH-SAW biosensor has great potential for a variety of applications, it has some limitations that need to be addressed: (i) The sd-LDL measured by the SH-SAW biosensor may not reflect the concentration of this fraction obtained by other assays, which needs to be investigated in future studies. (ii) The value of the SH-SAW biosensor must be reinforced in future studies with larger samples of patients and controls. (iii) More comparisons of the SH-SAW biosensor with other methods, especially ultracentrifuges, are necessary to validate whether the SH-SAW biosensor can measure the classical sd-LDL fraction. In this study, the filtration technique was easier and faster in obtaining real-time results than the centrifugation method because it correlated better with the results of the commercial sd-LDL-C assay.

We also demonstrated the SH-SAW assay system based on easy-to-use, highly reliable, sensitive, and reproduction-free measurements. The SH-SAW biosensor platform coated with anti-ApoB protein provides highly sensitive and rapid ApoB detection. This suggests an innovative approach to sd-LDL quantitation that has the potential to improve the speed and accuracy of measurements in patients with high CVD risk. Finally, the SH-SAW biosensor platform has the potential to be developed in more aspects of in vitro diagnostic biomarkers due to the preprocessing approach.

## 4. Materials and Methods

### 4.1. Subjects

Blood was collected from different subjects, including patients in Chang Gung Memorial Hospital and healthy volunteers from the Healthy Clinic in CGMH, in accordance with the guidelines approved by the institutional review board of Chang Gung Memorial Hospital in Taiwan (IRB No.202101722B0). Twenty-one subjects aged 25 to 55 years with a wide range of plasma lipid levels were enrolled. Whole blood samples were collected from all subjects after fasting for at least 8 h in pre-coated heparin-sodium tubes (SST II Advance Vacutainer, BD, Mississauga, ON, Canada) and stored at 4 °C. After centrifugation at 1000× *g* for 5 min at 4 °C, the supernatant plasma was harvested, separated, and stored at −80 °C until analyzed. To ensure the reliability of the study, all measurements, which differed from those obtained before the sampling, were performed.

### 4.2. NMR Analysis

NMR analysis was performed on Bruker Avance III HD 600 MHz spectrometers (Bruker Biospin GmbH, Rheinstetten, Germany) equipped with a TXI probes and the Bruker SampleJet automatic robot cooling system set to 6 °C. The 100 μL plasma samples supernatant was mixed with 75 mM pH 7.4 sodium phosphate (buffer in 1:1 ratio) and 200 μL were ceded into a 3 mm × 4 inch Bruker SampleJet NMR tube.

Lipoprotein analysis reports containing around 112 lipoprotein paramecium for each plasma sample were generated using the Bruker IVDr Lipoprotein Subclass Analysis (B.I.-LISA) method. This is completed by mathematically interrogating and quantifying the −CH2 (δ = 1.25 ppm) and −CH3 (δ = 0.8 ppm) peaks of the 1D spectrum from normalization to the Bruker QuantRef manager among Topspin by using a PLS-2 regression model [31].

### 4.3. SH-SAW Biosensor Chips Assay

Immunoassay reader and SH-SAW biosensor chips were provided by tst biomedical electronics Co., Ltd. (Taoyuan, Taiwan). SH-SAW biochip devices were produced in batches of 100–500 chips, with an insertion loss in air stipulation of between 21 and 22 dB. SH-SAW biochip dual-channel (reference and detection channel) devices were used with 5 µL sample for each biochip run. After removing the signal from the reference NAP channel, the change in phase shift attributable to sample antibody binding was assessed from the sample (detection) channel [32,33].

### 4.4. Establishment of the SH-S.A.W. 4PL Standard Regression Curve

Prior to sample testing, a quality control calibration curve was established using a 0, 24, 48, 105 and 148 mg/dL Apolipoprotein B Calibrator (Beckman Coulter, Ireland) to guarantee working biochip production for the batch. The ApoB calibrator was used as the standard. The standards were measured using SH-SAW biosensor chips and a 4PL standard curve was established; the equation for the 4PL standard curve is as follows:Y = d + (a − d)/[1 + (X/c)^b]
where a, b, c, and d are coefficients. Y is the SH-SAW biochip phase shift attribute, X is the concentration of Apolipoprotein B. The Apolipoprotein B levels were quantified in mg/dL.

### 4.5. Centrifugation Assay Procedure

It is known that divalent cations and polyanions precipitate apolipoprotein B [27], which contains lipoproteins. Based on this in Hirano et al. [28], the results were tested with different combinations of polyanion and divalent cations and compared with the reference method, ultracentrifugation. They were able to find the combination that showed the best correlation with the results obtained from ultracentrifugation. In this study, a precipitation reagent (0.1 mL) containing 150 U/mL of heparin sodium salt (Heparin Sodium CAS 2608411, Frankfurter, Merck, Germany) and 90 mmol/L of MgCl_2_ (MgCl_2_·6H_2_O CAS 7791-18-6 ACS grade, USB, Kachchh, Gujarat, India) were added to each plasma sample (0.1 mL), mixed, and incubated for 10 min at 37 °C. The samples were placed in an ice bath and allowed to stand for 15 min, then the precipitate was collected by centrifuging tool at 15,000 rpm for 15 min at 4 °C, Megafuge 16R (Thermo Fisher, Am Kalkberg, Germany). The precipitates were strongly packed into the wall of the Eppendorf tube, and the supernatants were clear. The supernatants were collected for the SAW ApoB biochip and performed the measurement of sd-LDL-ApoB. Plasma sd-LDL were also detected by sd-LDL- cholesterol “Seiken” assay (Denka Seiken Co. Ltd., Tokyo, Japan) on the Toshiba 200 FR automatic biochemical analyzer (TOSHIBA, Tokyo, Japan)

### 4.6. Filtration Assay Procedure

A schematic diagram of the centrifugation and filtration procedures for detection of ApoB concentrations in plasma is shown in Figure 7. This filtration method is performed as follows. The precipitation reagent (0.1 mL) containing 150 U/mL sodium salt and 90 mmol/L MgCl_2_ was added to each plasma sample (0.1 mL) and incubated for 10 min at 37 °C. We selected three filtration rating centrifugal filters, with filtration pore sizes following 300 KDa, 200 nm (Pall Corporation, Bloomfield Hills, MI, USA) and 100 nm (Merck, Darmstadt, Germany) for precipitate removal. Then, centrifugal filters (Merck, Darmstadt, Germany) were used for centrifuging at 5000× *g* for 1 min at room temperature. After three filtration effluents were collected, the more buoyant lipoproteins in sediment were removed, and measurement was performed using an SH-SAW ApoB biochip, NMR, and ApoB-100 ELSIA assay (Human ApoB-100 assay, IBL-America, Minneapolis, MN, USA). The total assay time for this precipitation and filtration procedure using an SH-SAW ApoB biochip was shortened to 11 min.

### 4.7. Statistical Analysis

In this study, all measurement results were expressed as means ± SD (standard deviation). Statistical analysis was conducted using Student’s *t*-test. *p*-values of less than 0.05 were considered statistically significant. Statistical analyses were performed using SPSS statistics 17 (SPSS, Chicago, IL, USA).

## 5. Conclusions

In conclusion, the present study shows that the quantification of sd-LDL-C can be shown more clearly by using filtration technology with the SH-SAW biosensor platform. Since the current modified method gives results that are consistent with those reported by the original method, this simple and cost-effective technique can be easily adopted even in small clinical laboratories for the rapid quantification of atherogenic sd-LDL-C with conventional cholesterol detection reagents. We have developed a novel and simple method to quantify sd-LDL in plasma using heparin-Mg^2+^ precipitation. This precipitation method is suitable for automated analyzers and allows for the rapid measurement of many samples. We plan to apply this simple method in a large sample to determine the prevalence of sd-LDL and its association with established risk factors for coronary heart disease to confirm the atherogenic potential of sd-LDL.

## Figures and Tables

**Figure 1 ijms-25-01044-f001:**
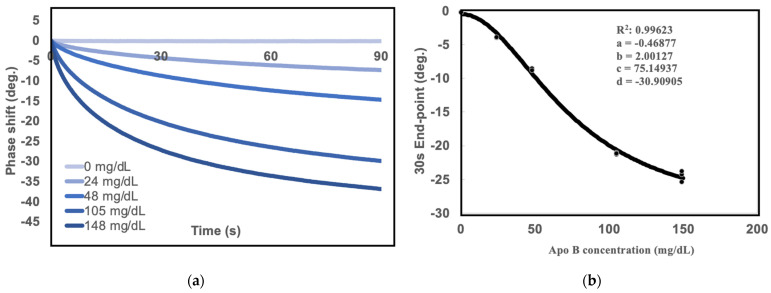
Phase shifts of the SH-SAW biosensor at different concentrations of ApoB samples and the 4PL curve of the SH-SAW biosensor. The purchased apolipoprotein B calibrator was diluted to different concentrations (24–148 mg/dL) and the samples were diluted to 20× and blank (PBS). Then, 5 μL sample drops were measured in the reaction area of the SAW chip and repeated three times. (**a**) Real-time curve of the measurements; (**b**) the 4PL fitting curve of the 30-s phase shift.

**Figure 2 ijms-25-01044-f002:**
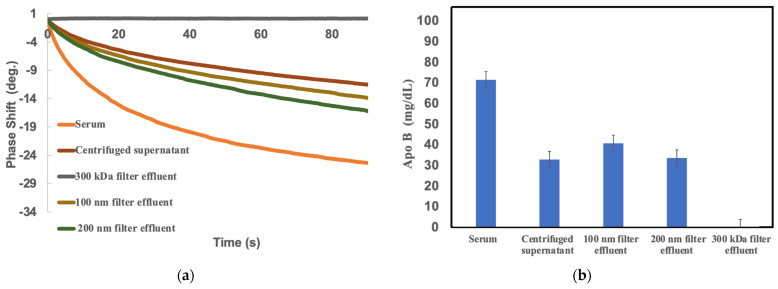
Phase shift measurements were performed on different samples using SH-SAW biosensors. Fresh serum was prepared for ApoB measurements. No precipitant was added to the serum. Samples, including centrifuged supernatant, 300 kDa filtered effluent, 100 nm filter effluent, and 200 nm filtered effluent, were spiked with precipitating heparin-Mg^2+^ precipitation reagents. After the incubation step, the supernatant samples were centrifuged and the ultrafiltrate was collected. The effluent samples were filtered through 300 kDa, 100 nm, and 200 nm filters and the effluent were collected. (**a**) Measured real-time curves; (**b**) SH-SAW ApoB measurements of different pretreatment serum samples.

**Figure 3 ijms-25-01044-f003:**
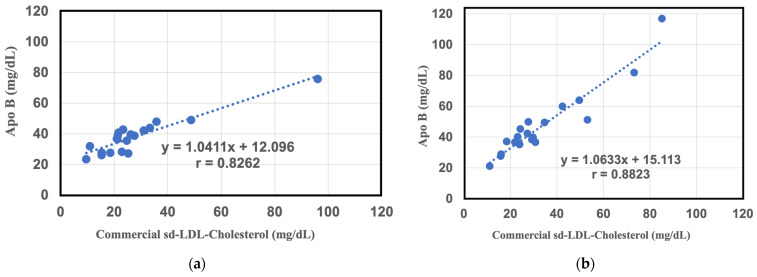
Comparative study between commercial sd-LD-C assay and SH-SAW ApoB biosensor measurements of different pre-treatment samples. (**a**) Correlation between commercial sd-LDL assay results and SH-SAW ApoB biosensor measurements of heparin-Mg^2+^ precipitated centrifuged supernatant (N = 20). (**b**) Correlation between commercial sd-LDL-C assay results and SH-SAW ApoB biosensor measurement on 100 nm pore size filtered effluent samples (N = 20).

**Figure 4 ijms-25-01044-f004:**
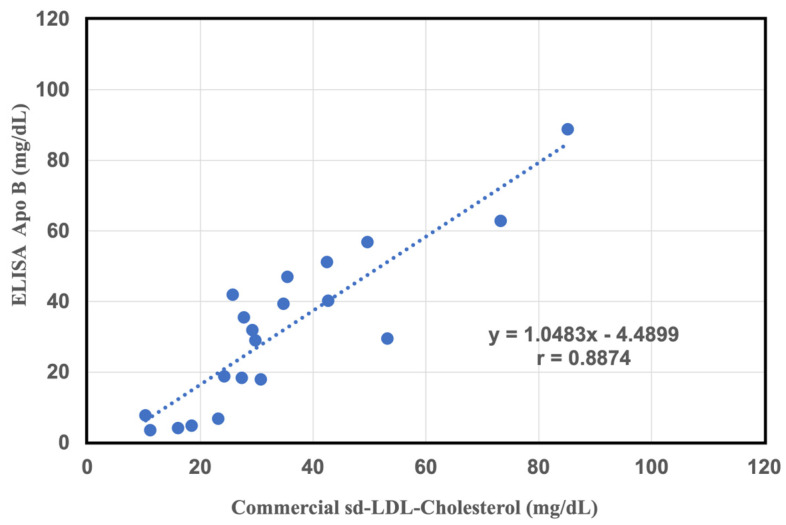
Correlation between the results of the commercial measurement of the sd-LDL-C assay and sd-LDL-ApoB measurement on 100 nm pore size filtered effluent samples using the ELISA ApoB-100 assay (N = 20).

**Figure 5 ijms-25-01044-f005:**
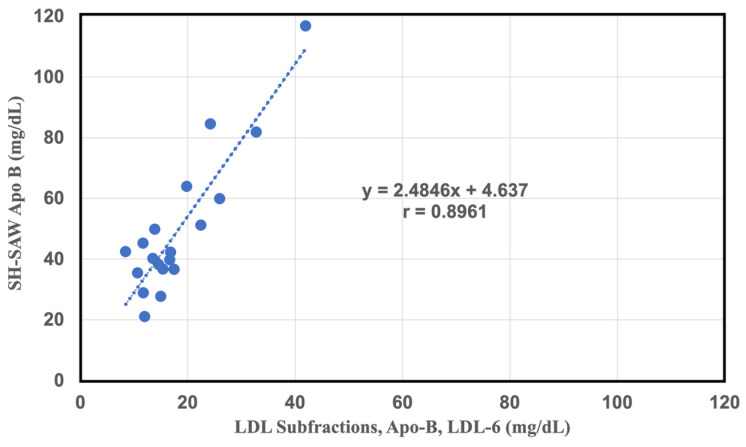
Correlation between 100 nm pore size filtration performed by the SH-SAW biochip in the sd-LDL-ApoB measurement of 20 individual plasma samples and flesh plasma performed by NMR spectra of the LDL subfractions, Apo-B, LDL-6 (N = 20). Correlation between 100 nm pore size filtration performed by the SH-SAW biochip in the sd-LDL-ApoB measurement of 20 individual plasma samples and fresh plasma performed by NMR spectra of the LDL subfractions, Apo-B, LDL-6 (N = 20).

**Figure 6 ijms-25-01044-f006:**
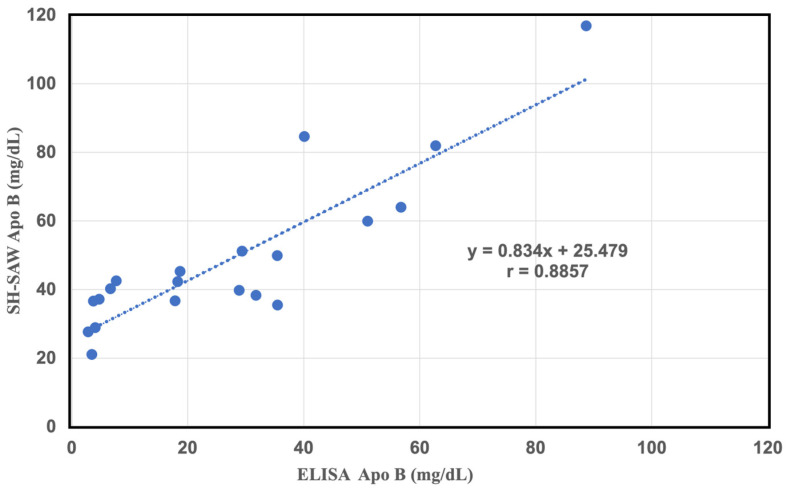
Correlation between the results of sd-LDL-ApoB concentrations measured in 100 nm pore size filtered effluent from 20 plasma samples using the SH-SAW biosensors and the ELISA ApoB-100 assay kit.

**Figure 7 ijms-25-01044-f007:**
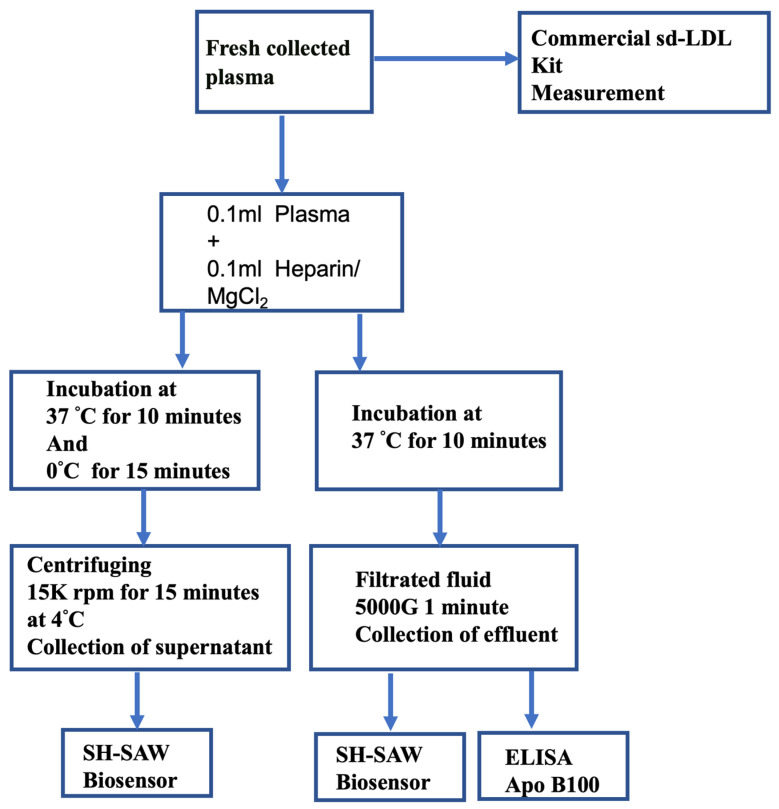
Schematic diagram of the centrifugation and filtration procedures for the detection ApoB in plasma.

**Table 1 ijms-25-01044-t001:** NMR analysis result of serum and different pretreatment method samples.

Samples	LDL Cholesterol (mg/dL)	Apo-B100 (mg/dL)	LDL-ApoB (mg/dL)	sd-LDL-ApoB (* LDL-6) (mg/dL)
Serum	76.9	63.2	49.7	14.8
Centrifuged Supernatant	9.0	33.0	18.4	2.9
300 KDa Effluent	0.0	29.6	15.9	0.0
100nm Effluent	7.4	29.7	13.7	1.0
200nm Effluent	13.5	36.2	19.6	3.3

* LDL-6 is Lipoprotein subclass of density of 1.044–1.063 g/mL.

## Data Availability

The datasets used and analyzed during the current study are available from the corresponding authors upon reasonable request.

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
