# Peer review of "New Advances in Rapid Pretreatment for Small Dense LDL Cholesterol Measurement Using Shear Horizontal Surface Acoustic Wave (SH-SAW) Technology"

_ijms, 2024, doi:10.3390/ijms25021044_

Round 1
Reviewer 1 Report (Previous Reviewer 3)
Comments and Suggestions for Authors
Tai-Hua Chou et al. report on a new method for quantitation of “small-dense low density lipoproteins (sd-LDL)”. This manuscript is a revision of a paper submitted earlier to ijms.
GENERAL COMMENTS
The manuscript contains in the Introduction and Discussion a great deal of textbook knowledge; since this is a pure methodological topic there is no need to explain in detail the features of atherosclerosis and stroke. The authors should concentrate rather on the characterization of apoB containing plasma lipoproteins in general and on sd-LDL in particular. Half a page of Introduction might be sufficient.
The authors point out in the Discussion correctly:”
“Although the distribution of LDL particles and their characteristics can be described by comparable nomenclature, it is debatable whether different methods define particles in the same way”.
This is a key point in defining sd-LDL. There are a lot of methods applied for quantitation of sd-LDL but no one knows exactly what they are measuring. Actually the experts in lipoprotein research understand in sd-LDL an apoB containing particle that possesses a higher buoyant density than regular LDL from healthy fasting individuals, floating by sequential ultracentrifugation at a density between 1.063 and 1.125. These lipoproteins are smaller as compared to regular LDL and are oxidized to a higher degree. Sd-LDL are mainly formed under hypertriglyceridemic conditions. Since they are more atherogenic than “normal” LDL simple and correct high-throughput methods are mostly welcome. It is therefore mandatory that any new method provides a thorough characterization of the analyte. This unfortunately is not fully true in this report.
Specific comments
1. Introduction, l.84 and Ref. 20: Ref. 20 is a review article reporting general features of sd-LDL assays; it is therefore not clear to what “ detergent based method for sd-LDL” is referred to.
2. ApoB calibrator from Beckman: Normally the industrial calibrators are specifically designed for the particular in-house assays; their exact composition and true concentration of the analyte in this calibrator is based on calculations and not known for the public. Thus it would be important for a new assay to prepare a purified fraction of sd-LDL according to standardized generally accepted methods and to use this specimen as a standard.
3. SH-SAW Biosensor: Since this is not a commonly used methodology the authors should describe it in much more detail: characterization of the antibody on the bio-chip, hardware for measuring signals, reproducibility, precision …. and so one.
4. Please state what kind of commercial assay for sd-LDL was used.
5. There are numerous figures shown displaying correlations. Actually correlations are useful for population studies, yet if it comes to individual patients a correlation of 0.88 does not mean much and is many times misleading. Looking for example at the graph in Fig. 4 and the points right from the regression line, there is one particular sample that shows approx. 30 mg/dl apoB by the ELISA and 55 by the commercial sd-LDL assay.
6. Also the slope of the regression line matters. In Fig. 5 the slope is far away from 45° and one wonders why the proposed SH-SAW method is propagated to measure sd-LDL as the values deviate quite largely from LDL-6 measured by NMR.
7. There are also numerous “outlayers” seen in the other graphs. It would help a lot if the authors would prepare a table showing all individuals numbers of measured concentrations from all 20 individuals comparing all assays that were applied. From the graphs it is hard to extract the real concentrations.
8. The conclusions made by the authors are based on the results from 20 samples of a very heterogenous population. Not only that the number of samples is small, it is not clear whether any of them suffered from diseases that typically express abnormal values of sd-LDL. In other words the authors should provide results from a minimum of 50 individuals with T2DM, metabolic syndrome and/or mild hypertriglyceridemia and compare the data from SH-SAW biosensor assaqy with that of MR and the commercial sd-LDL assay.
9. A few words on the cost of the SH-SAW Biosensor assay would be appropriate.
Comments on the Quality of English LanguageThe English is understadable - yet a correction by an English speaking person might improve it.
Author Response
as shown in the attached file

Reviewer 2 Report (New Reviewer)
Comments and Suggestions for Authors
This study aims to develop a novel SH-SAW-based technology to measure sd-LDL-ApoB in human plasma. It is very interesting and worth to try.
However, there are weaknesses in the manuscript. 1. SH-SAW and ELISA have similar results. What are the advantages for SH-SAW? 2. More details needed for SH-SAW describing. 3. In discussion: LDL usually associated with CVDs. Mention carcinogenicity of LDL seems weird. 4. What are the limitations for SH-SAW? This should be discussed in discussion. 5. All figures are blur and hard to read. 6. Where is t-test used in the manuscript?
Other minors:
Line 23: Is CAD or CVD?
Line 105: What is dense fraction of LDL?
Line 126, 137, 351, and Figure 1a: 148mg/dL and 149mg/dL not consistent.
Table1 needs top frame, and the font needs to be consistent.
Line 188: Pre-Pre: typo err?
Line 245-247: Not understandable, rewrite it please.
Line 257: flesh plasma or fresh plasma?
Line 274: What is lb-LDL?
Comments on the Quality of English LanguageEnglish is fine. Minor editing is needed.
Author Response
as shown in the attached file

Round 2
Reviewer 1 Report (Previous Reviewer 3)
Comments and Suggestions for Authors
While the manuscript has been improved considerably there are still some points that should be considered by the authors:
1. The introduction still needs to be streamlined focusing on the particular topic of this manuscript
2. In the Discussion, section "limitations" it should be stressed that i) sdLDL measured by the SH-SAW method might not reflect the concentration of this fraction obtained by other assays, and this needs to be studied in future research. ii) A much larger sample of patients and controls will have to be studied in future research to solidify the value of the SH-SAW method and iii) additional comparison of the SH-SAW method with other methods – in particular with the ultracentrifuge will be necessary to verify that SH-SAW in fact measures the classical sd-LDL fraction.
Author Response
Please see the attachment.

This manuscript is a resubmission of an earlier submission. The following is a list of the peer review reports and author responses from that submission.
Round 1
Reviewer 1 Report
Comments and Suggestions for Authors
I’ve read with attention the paper of Chou et al. that is potentially of interest. The background and aim of the study have been clearly defined. The methodology applied is overall correct, the results are reliable and adequately discussed. I’ve only some minor comments:
- The abstract should shorten the long introduction while enriching (if possibile with a couple of numeric data) the results section
- The statistics description seems a bit poor. A t-test only is reported (while it is not clear if the parameters were normally distribute), but the authors reports at least a regression line. This section should be detailed.
Comments on the Quality of English LanguageI’ve read with attention the paper of Chou et al. that is potentially of interest. The background and aim of the study have been clearly defined. The methodology applied is overall correct, the results are reliable and adequately discussed. I’ve only some minor comments:
- The abstract should shorten the long introduction while enriching (if possibile with a couple of numeric data) the results section
- The statistics description seems a bit poor. A t-test only is reported (while it is not clear if the parameters were normally distribute), but the authors reports at least a regression line. This section should be detailed.
Author Response
I’ve read with attention the paper of Chou et al. that is potentially of interest. The background and aim of the study have been clearly defined. The methodology applied is overall correct, the results are reliable and adequately discussed. I’ve only some minor comments:
- The abstract should shorten the long introduction while enriching (if possibile with a couple of numeric data) the results section
Response: Thanks for the reviewer’s point. We have modified the abstract (including some results). Please refer to the revised manuscript.
- The statistics description seems a bit poor. A t-test only is reported (while it is not clear if the parameters were normally distribute), but the authors reports at least a regression line. This section should be detailed.
Response: Thanks for review’s recommendation. In fact, we didn’t use t-test in this manuscript. In the comparative study, we use the correlation coefficient to check the correlation between the two different sd-LDL assays. Statistical methods and results were modified and adjusted.
Reviewer 2 Report
Comments and Suggestions for Authors
-The introduction is quite long, confusing and repetitive in some parts. The reference to the CV disease management in the low and middle income countries does not strictly relate to the objective of th study. Authors should better focus on the description of the state of the art that is necessary to understand the novelty of their work.
-Lines 299-300: I did not get this point. Explain more clearly
-line 307: I did not understand the role of lipoprotein paramecium
-talking abouth the filtration method, it is not clear to me why some filters are indicated with the pore size (100-200 nm) and an other with MW (300kDa). How is the comparison possible?
-I do not agree with the statement on line 169. Data from 200nm filters seem to be more similar to the reference. Authors has to support their consideration. Was the difference between samples analyzed by a statistical point of view? Reading this version of the paper, I think that the following test should have been performed using 200nm filter.
-In the discussion a long sentence is repeated. Authors should pay more attention in the final revision of the text
Comments on the Quality of English LanguageNot fluent. Many typos all along th text.
Author Response
-The introduction is quite long, confusing and repetitive in some parts. The reference to the CV disease management in the low and middle income countries does not strictly relate to the objective of the study. Authors should better focus on the description of the state of the art that is necessary to understand the novelty of their work.
Response: We thank the reviewers for their comments. Yes, in the revised manuscript we have made some changes to the novelty of this study.
-Lines 299-300: I did not get this point. Explain more clearly
Response: The lipoprotein subclass analysis was performed by NMR spectrometer. Data was obtained on Bruker Avance III HD 600 MHz spectrometers (Bruker Biospin GmbH, Rheinstetten, Germany) equipped with a TXI probes and the Bruker SampleJet automated robotic cooling system set to 6 °C.”
-line 307: I did not understand the role of lipoprotein paramecium
Response: It’s typing error. “lipoprotein parameters” was correct sentence.
-talking about the filtration method, it is not clear to me why some filters are indicated with the pore size (100-200 nm) and another with MW (300kDa). How is the comparison possible?
Response: Thanks for the reviewer’s point. This filtration rating is derived from the filter manufacturer's product specifications. We can refer to the following article, which reveals the range from microscopic rating scale to MW rating scale.
Pearce, G. (2007). Introduction to membranes: Filtration for water and wastewater treatment. Filtration & Separation, 44(2), 24-27. https://doi.org/https://doi.org/10.1016/S0015-1882(07)70052-6
-I do not agree with the statement on line 169. Data from 200nm filters seem to be more similar to the reference. Authors has to support their consideration. Was the difference between samples analyzed by a statistical point of view? Reading this version of the paper, I think that the following test should have been performed using 200nm filter.
Response: We thank the reviewers for their comments. In our study, the 200-nm filter leaked more ApoB in its effluent. We add the figure 1b to support this finding. And from NMR measurement Table 1 showed the 200-nm filter detected more LDL-ApoB than the 100-nm effluent. sd-LDL-ApoB in serum was analyzed more closely to the 100-nm effluent.
-In the discussion a long sentence is repeated. Authors should pay more attention in the final revision of the text
Response: Thanks for the reviewer’s point. We have revised the sentences. Please refer to the revised manuscript.
Reviewer 3 Report
Comments and Suggestions for Authors
Tai-Hua Chou describe a new biosensor method for measurement of small-dense (sd-) LDL in human plasma. It is concluded that this method might be suitable for routine measurements in the clinical laboratory.
GENNERAL COMMENT
The authors are right that sd-LDL is an important parameter to assess the CVD risk in a given population. While the propagated method might be well appropriate for the proposed purpose, there is not enough evidence that it measures truly only sd-LDL and NOT some other apoB containing components of plasma. In other words the authors need to verify that what they measure is comparable with the conventional sd-LDL fraction that was studied in numerous reports to demonstrate their correlation with CVD and heart diseases.
SPECIFIC COMMENTS
1. The Introduction is not focused appropriately and in particular the first 2 § are not pertinent to the aim of this study.
2. Results, 2.1. It is not clear what kind of ApoB calibrator was used for constructing the standard curves. Was that calibrator pure sd-LDL or was it just any LDL fraction, or anything else. If it is not genuine sd-LDL it is probably not really suited for this study. The best would actually be that the authors prepare pure sd-LDL freshly and before using to characterize it unequivocally.
3. Also, it is said that “the sample were diluted into twenty times “ – What does this mean? The authors should mention how many standards they prepared and what the actual concentration of each sd-LDL in these standards had been.
4. In Fig.2 the actual measured values for the different concentrations should be shown (the measurement points for constructing the curves).
5. Methods: The description of the methods is more than superficial. It is not clear how the samples were prepared before putting them on the SH-SAW apoB biochip. If I understand correctly in a first step the apoB containing lipoproteins were precipitated with a heparin-magnesium reagent and after centrifugation, the supernatants were then filtered. There is actually no proof that this methods just prepares the bulk of sd-LDL and nothing is lost during the different steps. In order to understand what the exact procetre was, a flow-sheet with an exact description should be provided.
6. Results 2.3: Which commercial sd-LDL-C kit was used? Please give the exact company and batch.
7. NMR-Method: There is actually a NMR-method that directly measures sd-LDL in whole plasma. Why did the authors not use this methods for comparison in the “large study group”?
Taking together, the authors must describe their method including all used reagents and batches in detail and in a way that the reader might be able to reproduce the work. Also it is not clear that the comparison with a commercial kit in fact gives a correct answer. The golden standard for sd-LDL is to separate it by ultracentrifugation making sure that it is not contaminated with other apoB containing lipoproteins such as Lp(a), followed by measuring LDL-C in the sd-LDL fraction.
Comments on the Quality of English Language
The English needs intensive editing
Author Response
GENNERAL COMMENT
The authors are right that sd-LDL is an important parameter to assess the CVD risk in a given population. While the propagated method might be well appropriate for the proposed purpose, there is not enough evidence that it measures truly only sd-LDL and NOT some other apoB containing components of plasma. In other words the authors need to verify that what they measure is comparable with the conventional sd-LDL fraction that was studied in numerous reports to demonstrate their correlation with CVD and heart diseases.
Response: Thanks to the reviewer’s point, according to the results in Figure 2, the ApoB measurement by SH-SAW is significantly corrected with the commercial sdLDL kit indicating that the conventional sd-LDL fraction is relevant to our ApoB measurement. We strongly believe that we can use the SH-SAW ApoB sensor, together with a simple filtering process, which can be applied as a clinical tool.
SPECIFIC COMMENTS
- The Introduction is not focused appropriately and in particular the first 2 sections?? are not pertinent to the aim of this study.
Response: Thanks for the reviewer’s point, we have modified the aim of this study as shown in the revised manuscript.
- Results, 2.1. It is not clear what kind of ApoB calibrator was used for constructing the standard curves. Was that calibrator pure sd-LDL or was it just any LDL fraction, or anything else. If it is not genuine sd-LDL it is probably not really suited for this study. The best would actually be that the authors prepare pure sd-LDL freshly and before using to characterize it unequivocally.
Response: Thanks for the reviewer’s point. In this study, we use purified ApoB protein as a calibrator of SH-SAW biosensors. The ApoB was the major apolipoprotein in lipoprotein, especially in VLDL and LDL. After precipitation and filtration, the concentration of sd-LDL could be defined by detecting ApoB. This is the reason we use ApoB as a calibrator to estimate the sd-LDL-apoB concentration of the processed plasma sample.
Reference: (number#28, in the revised manuscript)
Hirano, T., Ito, Y., Saegusa, H., & Yoshino, G. (2003). A novel and simple method for quantification of small, dense LDL. J Lipid Res, 44(11), 2193-2201. https://doi.org/10.1194/jlr.D300007-JLR200
- Also, it is said that “the sample were diluted into twenty times“ – What does this mean? The authors should mention how many standards they prepared and what the actual concentration of each sd-LDL in these standards had been.
Response: Thanks for the reviewer for pointing out this section. In this study, we found that the samples need to be diluted up to 20-fold and will result in better 4PL regression in SH-SAW biosensor. As our answer to question #2, precipitation and filtration, the concentration of sd-LDL can be defined by detecting ApoB. This is why we used ApoB as a calibrator to estimate the sd-LDL-apoB concentration in treated plasma samples.
- In Fig.2 the actual measured values for the different concentrations should be shown (the measurement points for constructing the curves)
Response: We thank the reviewers for their views. In this experiment, we attempted to use the high sensitivity of SH-SAW to initially screen for the best filtration grade to remove the precipitates. Our aim was not to obtain accurate apolipoprotein concentrations from SH-SAW. We used NMR as a tool to show the apolipoprotein concentration for each test, see Table 1.
- Methods: The description of the methods is more than superficial. It is not clear how the samples were prepared before putting them on the SH-SAW apoB biochip. If I understand correctly in a first step the apoB containing lipoproteins were precipitated with a heparin-magnesium reagent and after centrifugation, the supernatants were then filtered. There is actually no proof that this methods just prepares the bulk of sd-LDL and nothing is lost during the different steps. In order to understand what the exact protocol was, a flow-sheet (flowchat) with an exact description should be provided.
Response: Thanks to the reviewers for pointing out this section. In our study, we wanted to demonstrate that it is not necessary to use prolonged high-speed centrifugation (15k rpm, 15 min) to remove non-LDL precipitates. In Figure 2.b, we did demonstrate that a 100 nm filter also removes non-LDL precipitates and shows better correlation with commercial sd-LDL-cholesterol assays.
- Results 2.3: Which commercial sd-LDL-C kit was used? Please give the exact company and batch.
Response: We used the commercial name “s LDL-EX “SEIKEN” assay for sd-LDL measurement, the batch Lot# 601121.
- NMR-Method: There is actually a NMR-method that directly measures sd-LDL in whole plasma. Why did the authors not use this methods for comparison in the “large study group”?
Response: Thanks for the reviewer’s comments. In our study, we sought to identify a new pretreatment measurement method to detect sd-LDL in a rapid and reliable way. Our goal was to compare current commercial clinical methods (sd-LDL-cholesterol "Seiken" assay) to laboratory tools similar to NMR or ultracentrifugation systems.
Taking together, the authors must describe their method including all used reagents and batches in detail and in a way that the reader might be able to reproduce the work. Also it is not clear that the comparison with a commercial kit in fact gives a correct answer. The golden standard for sd-LDL is to separate it by ultracentrifugation making sure that it is not contaminated with other apoB containing lipoproteins such as Lp(a), followed by measuring LDL-C in the sd-LDL fraction.
Response: Thanks for the reviewer’s comments. Ultracentrifugation is the gold standard for sd-LDL, but it is too laboratory and complex a method for clinical application. The LDL-EX “SEIKEN” is now a commercially available kit and approved by the FDA. We compared this commercial kit with our newly developed pretreatment method with SH-SAW biosensor. In our study, the ApoB of SH-SAW results showed a good correlation with the commercial kits. The batches of the commercial kit was showed in the revised manuscript.